# New Construction of Asynchronous Channel Hopping Sequences in Cognitive Radio Networks

**DOI:** 10.3390/e25101473

**Published:** 2023-10-22

**Authors:** Yaoxuan Wang, Xianhua Niu, Chao Qi, Zhihang He, Bosen Zeng

**Affiliations:** 1Department of Computer and Software Engineering, Xihua University, Chengdu 610039, China; adalyssa@163.com (Y.W.); chaoqi@mail.xhu.edu.cn (C.Q.); hkaitlinruby417@gmail.com (Z.H.); 2Chengdu Institute of Computer Applications, Chinese Academy of Sciences, Chengdu 610041, China; zengbosen19@mails.ucas.ac.cn

**Keywords:** cognitive radio networks, channel-hopping sequence, asynchronous

## Abstract

The channel-hopping-based rendezvous is essential to alleviate the problem of under-utilization and scarcity of the spectrum in cognitive radio networks. It dynamically allows unlicensed secondary users to schedule rendezvous channels using the assigned hopping sequence to guarantee the self-organization property in a limited time. In this paper, we use the interleaving technique to cleverly construct a set of asynchronous channel-hopping sequences consisting of *d* sequences of period $xN^{2}$ with flexible parameters, which can generate sequences of different lengths. By this advantage, the new designed CHSs can be used to adapt to the demands of various communication scenarios. Furthermore, we focus on the improved maximum-time-to-rendezvous and maximum-first-time-to-rendezvous performance of the new construction compared to the prior research at the same sequence length. The new channel-hopping sequences ensure that rendezvous occurs between any two sequences and the rendezvous times are random and unpredictable when using licensed channels under asynchronous access, although the full degree-of-rendezvous is not satisfied. Our simulation results show that the new construction is more balanced and unpredictable between the maximum-time-to-rendezvous and the mean and variance of time-to-rendezvous.

## 1. Introduction

Cognitive radio is one proposed solution to address the issue of limited spectrum resources in wireless communication systems and enhance spectrum utilization. The fundamental concept is to enable wireless communication devices to locate and utilize “spectrum holes” intelligently [1,2]. In cognitive radio networks, the communication between secondary users (SUs) requires a rendezvous point to establish an initial communication link by acquiring a common control channel through a spectrum database [3]. However, this centralized approach cannot avoid conflicts and control channel bottlenecks in a multi-channel, multi-user scenario [4], so it is more challenging to implement a rendezvous based on a distributed approach [5]. Figure 1 depicts the rendezvous between SUs through a distributed approach. SUs communicate in pairs, and when the primary user (PU) occupies the licensed channel, the SUs experience interference. Thus, the SUs rendezvous fails. The PU shares the licensed channels with the unlicensed SUs. When two SUs jump to the same licensed channel, rendezvous communication can be achieved. In most wireless communication technologies, the channel-hopping sequence (CHS) is a typical technique to achieve distribution.

CHS methods are generally classified by different symmetric/asymmetric roles and synchronous/asynchronous modes [6]. The symmetric role usually refers to the flexibility of SUs to act as a sender and a receiver with only one CHS. For example, a group of radio intercoms communicates, where the user’s intercom acts as the receiver. When a user sends a message over the walkie-talkie, that user’s walkie-talkie acts as the sender at that time. When the other users reply to the news, the user’s walkie-talkie picks it up and converts it to an audio signal. In contrast, the asymmetric role requires two separate CHSs to be assigned to each SU in advance to serve as the receiver [7]. When there are many SUs, it is evident that the symmetric role is more beneficial than the asymmetric one, and the symmetric part can be appropriately converted to the asymmetric position [6].

In the synchronization model, the SUs must operate within strict time constraints to ensure they send and receive information in the correct channel. Therefore, global synchronization is critical because all SUs must operate on the same schedule. However, in the asynchronous model, SUs can operate according to their propagation times (i.e., self-organization). They can be partially synchronized with other SUs to handle propagation delays and timing issues more flexibly [8]. Figure 2 shows three users that each have an independent CHS and communicate two-by-two at different time delays. Users can communicate with each other as long as they hop into the same licensed channel simultaneously. Due to the advantages mentioned above, therefore, asynchronous symmetric CHSs are investigated in this paper.

In order to address issues such as a higher parameter demand and maximum-time-to-rendezvous (MTTR) and maximum-first-time-to-rendezvous (MFTTR) performance, this paper focuses on a set of CHSs with new parameters that can be obtained with more CHSs of different lengths suitable for different scenarios. These sequences have shorter MTTR and MFTTR when equipped with the same parameters. The article’s structure is shown below: Section 2 briefly compares existing CHSs and summarizes the contribution. Section 3 describes in detail the natural number of field CHSs with new parameters. In Section 4, a comprehensive analysis of the characteristics of the new and existing constructions is provided by the simulation results. Finally, Section 5 summarizes and generalizes the previous studies and results.

## 2. Related Work and Contribution

Referring to the previous work, in the literature [9,10,11], the CHSs are designed under full degree-of-rendezvous(DoR) constraint. While in the literature [12,13,14,15,16], the CHSs are designed focusing on good MTTR and MFTTR performance without full DoR constraint, these CHSs are also applicable in many communication scenarios.

Liu [12] proposed a blind rendezvous algorithm, which employs a jump–stay strategy. In this algorithm, unavailable channels are randomly replaced with available channels to increase the chances of rendezvous. It is suitable for scenarios with multiple users and multiple hops. Bian [13] proposed a novel distributed protocol based on “neighbourhood diversity”. This protocol enables highly differentiated node encounters by selecting the optimal encounter channel and spectrum. Chuang [14] developed a method to achieve rendezvous in a cognitive radio network. The method involves alternating between frequency hopping and waiting algorithms, enabling nodes to efficiently locate available communication channels without needing global time synchronization. Sheu [9] proposed an asynchronous Quorum-based blind rendezvous scheme to achieve coordination between nodes by nodes communicating on different channels and time slots. Yang [10] has proposed a rendezvous algorithm based on Disjoint Set Cover (DSC) called Disjoint Set Cover Rendezvous (DSCR) to overcome limitations such as clock synchronization. DSC is utilized to adjust the arrangement of access channels, enabling users to quickly and efficiently rendezvous on all available channels. New asynchronous symmetric CHS were constructed, and their properties of channel overlap, even channel usage and pairwise shift invariance were investigated [15]. Chang [16] provided a new solution to improve the efficiency and effectiveness of rendezvous in cognitive radio networks by closing the theoretical gap in the multichannel rendezvous problem and introducing the IDEAL-CH sequence with an asymptotic approximation ratio of 2. Yang [11] developed a set of asynchronous CHSs based on IDs in a homogeneous setting. Their approach involved utilizing matrices and prime numbers to create algebraic algorithms.

Generally speaking, with the development of wireless networks and communication scenarios, the design of CHSs has the following two demands:

1. Due to the variety of wireless network topologies, more CHSs with different lengths are needed to support the communications. The existing work designed CHSs of length *L* as a function of $N^{2}$ [9,12,13,14] or prime *p* [10,11,15,16].

2. Since shorter MTTR and MFTTR can improve the system throughput and spectrum utilization, new CHSs with short MTTR and MFTTR are demanded. The MTTR and MFTTR of [9,10,14] are shown in columns 2 and 3 of Table 1.

To meet the demand of CHSs in various communication scenarios, this paper focuses on using the interleaving to cleverly design CHSs consisting of *d* sequences of period $xN^{2}$ with new parameters, which can generate sequences of different lengths and achieve shorter MTTR and MFTTR. The newly designed CHSs can be applied to various communication scenarios with complex network topologies. Using the designed new set of CHSs, the new construction can generate shorter sequences for better application where the network topology is changing rapidly; conversely, the new construction can also generate longer sequences for duration communication, thus, making communication safer and more reliable. The main contributions of this paper are:

(1) The new construction extends the period-length results of the existing constructions [9,11,13,15,16]. When *N* is a natural number, the period results from [9,13] are a special case of the results of our Theorem 2. When *N* = prime *p*, the period results from [11,15,16] are special cases of our Corollary 1 result.

(2) Compared to the previous studies without full DoR [12,13,14,15,16], the novel construction places its emphasis on achieving improved MTTR and MFTTR performance at identical sequence lengths when compared to the existing literature, although the full DoR is not satisfied. For example, when x=6 and l=1, our constructed sequence has MTTR = $3N^{2}$ and MFTTR = $3N^{2}$; whereas in [13], MTTR = $6N^{2}-N+1$ and MFTTR = $6N^{2}$. When N=p and x=2, the MTTR = 2p−1 of our constructed sequence is 1 smaller than in [11], while the MFTTR is equal. Thus, the new CHS has smaller MTTR and MFTTR, but longer throughput.

## 3. Construction of Channel Hopping Sequence Sets with New Parameters Based on Natural Numbers

We created CHS sets by a unique interleaving construction in the natural number field. This method uses natural number sequences to construct $xN^{2}$ CHSs with new parameters, called NPCH. It is guaranteed that any two shifted copies of the CHS set will have a mutual rendezvous with an integer N≥3. The $xN^{2}$ CHS satisfy an irregular time-to-rendezvous (TTR) pattern, which makes the new CHS set unpredictable and unreliable during communication rendezvous. This design ensures reliability. Table 1 in Section 2 shows that this construction has the shortest MTTR and MFTTR under the same conditions.

### 3.1. NPCH Construction


**Construction 1.**
*
**Construction of CHS sets**
*


Step 1: Choose an integer N≥3 and $\gamma \in Z_{N}^{*}$ with the largest multiplicative order. Let ord(γ)=d with maximum value φ(N), where φ is the Euler function. Without loss of generality, d=φ(N).

Step 2: For any λ∈[0,d−1], the CHS $S^{\lambda }$ can be written as:$$S^{\lambda }=\left[\begin{array}{cccc} s_{0,0}^{\lambda } & s_{0,1}^{\lambda } & \cdots & s_{0,N-1}^{\lambda }\\ s_{1,0}^{\lambda } & s_{1,1}^{\lambda } & \cdots & s_{1,N-1}^{\lambda }\\ \vdots & \vdots & \cdots & \vdots \\ s_{N-1,0}^{\lambda } & s_{N-1,1}^{\lambda } & \cdots & s_{N-1,N-1}^{\lambda } \end{array}\right]$$
where
(1)$$s_{\alpha ,\beta }^{\lambda }=\gamma^{\lambda }\left(\alpha +\beta +\lambda \right)\left(\mathrm{mod}N\right)$$
α,β∈[0,N−1].

Step 3: For any λ∈[0,d−1], the CHS $M^{\lambda }$ can be obtained by stringing the matrix $S^{\lambda }$*x* times, which can be expressed as follows:$$\underset{x}{\underbrace{\left[S^{\lambda }|\cdots |S^{\lambda }\right]}}$$

The specific algorithm is described in Algorithm 1.
**Algorithm 1:** Algorithm to generate CHS set with new parameters.**Input:** Integer N≥3, *x***Output:** Channel-hopping sequences set $M=\{M^{\lambda }|\lambda \in \left[0,d-1\right]\}$ 1:γ← max(find_primitive_roots(*N*)); 2:set λ={0,1,...,d−1}, d=φ(N); // φ is the Euler function; 3:S[]← generate_matrix(*N*, λ, γ); 4:**for** i=0 to *x* **do** 5:   **if** i==0 **then** 6:     $M^{\lambda }$←S[]; 7:   **else** 8:     $M^{\lambda }$←$\left[M^{\lambda }|S\right]$; 9:   **end if** 10:**end for** 11:**
return** $M^{\lambda }$← expand $M^{\lambda }$ line by line.

### 3.2. Constructive Analysis

Since the new construction rendezvous is generated by the $xN^{2}$ CHS “horizontal” and “vertical” shifts, Theorem 1 below shows that channel rendezvous is guaranteed in any two shifted CHSs.

**Theorem** **1.**
*The set M in Construction 1 is CHS set with d sequences of period $xN^{2}$, and there is at least one rendezvous between any two CHSs in the set at any time delay τ for any N≥3.*


**Proof.** It is easily checked that the CHS set M in Construction 1 is with *d* sequences of length $L=xN^{2}$ for any N≥3. Next, we prove that the rendezvous point of any two sequences in M is at least one.For any $\lambda_{1}\in \left[0,d-1\right]$, the CHS $M^{\lambda_{1}}$ can be expressed by the N×xN matrix as follows:
$$\left[\begin{array}{ccccccc} s_{0,0}^{\lambda_{1}} & \cdots & s_{0,N-1}^{\lambda_{1}} & \cdots & s_{0,0}^{\lambda_{1}} & \cdots & s_{0,N-1}^{\lambda_{1}}\\ s_{1,0}^{\lambda_{1}} & \cdots & s_{1,N-1}^{\lambda_{1}} & \cdots & s_{1,0}^{\lambda_{!}} & \cdots & s_{1,N-1}^{\lambda_{1}}\\ \vdots & \cdots & \vdots & \cdots & \vdots & \cdots & \vdots \\ s_{N-1,0}^{\lambda_{1}} & \cdots & s_{N-1,N-1}^{\lambda_{1}} & \cdots & s_{N-1,0}^{\lambda_{!}} & \cdots & s_{N-1,N-1}^{\lambda_{1}} \end{array}\right]$$For any time delay τ=xNv+h,0≤h≤xN−1,0≤v≤N−1, the shift sequence $R^{\tau }\left(M^{\lambda_{2}}\right)$ can be expressed by the N×xN matrix as follows:
$$\underset{a\;N\times N\;\mathrm{matrixs}}{\underbrace{\left[\begin{array}{ccccccccc} s_{N-v-1,N-b}^{\lambda_{2}} & \cdots & s_{N-v-1,N-1}^{\lambda_{2}} & s_{N-v-1,0}^{\lambda_{2}} & \cdots & s_{N-v-1,N-b-1}^{\lambda_{2}} & \cdots & \cdots & \\ s_{N-v,N-b}^{\lambda_{2}} & \cdots & s_{N-v,N-1}^{\lambda_{2}} & s_{N-v,0}^{\lambda_{2}} & \cdots & s_{N-v,N-b-1}^{\lambda_{2}} & \cdots & \cdots & \\ \vdots & \cdots & \vdots & \vdots & \cdots & \vdots & \cdots & \cdots & \\ s_{0,N-b}^{\lambda_{2}} & \cdots & s_{0,N-1}^{\lambda_{2}} & s_{0,0}^{\lambda_{2}} & \cdots & s_{0,N-b-1}^{\lambda_{2}} & \cdots & \cdots & \\ \vdots & \cdots & \vdots & \vdots & \cdots & \vdots & \cdots & \cdots & \\ s_{N-v-2,N-b}^{\lambda_{2}} & \cdots & s_{N-v-2,N-1}^{\lambda_{2}} & s_{N-v-2,0}^{\lambda_{2}} & \cdots & s_{N-v-2,N-b-1}^{\lambda_{2}} & \cdots & \cdots & \end{array}\right.}}$$
$$\left.\begin{array}{ccccccccccccc} & s_{N-v-1,N-b}^{\lambda_{2}} & \cdots & s_{N-v-1,N-1}^{\lambda_{2}} & s_{N-v,0}^{\lambda_{2}} & \cdots & s_{N-v,N-b-1}^{\lambda_{2}} & \cdots & s_{N-v,N-b}^{\lambda_{2}} & \cdots & s_{N-v,0}^{\lambda_{2}} & \cdots & s_{N-v,N-b-1}^{\lambda_{2}}\\ & s_{N-v,N-b}^{\lambda_{2}} & \cdots & s_{N-v,N-1}^{\lambda_{2}} & s_{N-v+1,0}^{\lambda_{2}} & \cdots & s_{N-v+1,N-b-1}^{\lambda_{2}} & \cdots & s_{N-v+1,N-b}^{\lambda_{2}} & \cdots & s_{N-v+1,0}^{\lambda_{2}} & \cdots & s_{N-v+1,N-b-1}^{\lambda_{2}}\\ & \vdots & \cdots & \vdots & \vdots & \cdots & \vdots & \cdots & \vdots & \cdots & \vdots & \cdots & \vdots \\ & s_{0,N-b}^{\lambda_{2}} & \cdots & s_{0,N-1}^{\lambda_{2}} & s_{1,0}^{\lambda_{2}} & \cdots & s_{1,N-b-1}^{\lambda_{2}} & \cdots & s_{1,N-b}^{\lambda_{2}} & \cdots & s_{1,0}^{\lambda_{2}} & \cdots & s_{1,N-b-1}^{\lambda_{2}}\\ & \vdots & \cdots & \vdots & \vdots & \cdots & \vdots & \cdots & \vdots & \cdots & \vdots & \cdots & \vdots \\ & s_{N-v-2,N-b}^{\lambda_{2}} & \cdots & s_{N-v-2,N-1}^{\lambda_{2}} & s_{N-v-1,0}^{\lambda_{2}} & \cdots & s_{N-v-1,N-b-1}^{\lambda_{2}} & \cdots & s_{N-v-1,N-b}^{\lambda_{2}} & \cdots & s_{N-v-1,0}^{\lambda_{2}} & \cdots & s_{N-v-1,N-b-1}^{\lambda_{2}} \end{array}\right]$$
where $\lambda_{1}\neq \lambda_{2}\in \left[0,d-1\right]$, b=hmodN, and *a* denotes a total of $\left\lfloor \frac{h}{N}\right\rfloor$ identical matrices.
The rendezvous number of any two sequences $M^{\lambda_{1}}$ and $M^{\lambda_{2}}$ under time-delay τ can be defined as follows:
(2)$$H_{M^{\lambda_{1}},M^{\lambda_{2}}}\left(\tau \right)=\sum_{\beta =0}^{xN-1}\left(\sum_{\alpha =0}^{xN-1}h\left(s_{\alpha ,\beta }^{\lambda_{1}},s_{\alpha +v,<\beta +h{>}_{N}}^{\lambda_{2}}\right)\right),\tau =xNv+h\left(0\leq h\leq xN-1,0\leq v\leq N-1\right)$$
where h(x,y)=1 if x=y, and h(x,y)=0; otherwise, $<x{>}_{y}$ is denoted as the least non-negative residue of *x* modulo *y* for two positive integers *x* and *y*.For any given CHS set M, for all τ∈[0,xN−1], the minimum number of rendezvous $H_{min}\left(M\right)$ is defined as follows:
(3)$$H_{min}\left(M\right)=\underset{0\leq \tau \leq xN^{2}-1}{min}\left\{H_{M^{\lambda_{1}},M^{\lambda_{2}}}\left(\tau \right):M^{\lambda_{1}},M^{\lambda_{2}}\in M,\lambda_{1}\neq \lambda_{2}\right\}$$By the construction of $s_{\alpha ,\beta }^{\lambda }$ in Equation (Equation 1), for any $\beta^{*}\in \left[0,xN-1\right]$, point $s_{\alpha ,\beta^{*}}^{\lambda_{1}}$ and point $s_{\alpha +v,<\beta^{*}+h{>}_{N}}^{\lambda_{2}}$ for the rendezvous are satisfied:
(4)$$\gamma^{\lambda_{1}}\left(\alpha +\beta^{*}+\lambda_{1}\right)\equiv \gamma^{\lambda_{2}}\left(\alpha \pm h+\beta^{*}\pm v+\lambda_{2}\right)\left(\mathrm{mod}N\right)$$Since $\gamma^{\lambda_{1}}\neq \gamma^{\lambda_{2}}$, Equation (Equation 3) can be simplified by operating in the finite field *N*:
(5)$$\gamma^{\lambda_{1}}i\equiv \gamma^{\lambda_{2}}j\left(\mathrm{mod}N\right)$$
where i,j∈[0,N−1].Clearly, Equation (Equation 5) holds if i=j=0. It indicates
$$\sum_{\alpha =0}^{N-1}h\left(s_{\alpha ,\beta^{*}}^{\lambda_{1}},s_{\alpha +v,<\beta^{*}+h{>}_{N}}^{\lambda_{2}}\right)\geq 1$$By Equations (2) and (3) for any $\beta^{*}\in \left[0,xN-1\right]$, we have
(6)$$H_{min}\left(M\right)\geq 1$$It is shown that there is at least one rendezvous between any two shift sequences.When *N* can be decomposed into $N=q_{1}q_{2}...q_{n}$, where $q_{1},q_{2},...,q_{u},...,q_{n}$ are all prime factors of *N*, if $\gamma^{\lambda_{1}}-\gamma^{\lambda_{2}}=\frac{N}{q_{u}}$ holds, we have:
$$q_{u}\left(\gamma^{\lambda_{1}}-\gamma^{\lambda_{2}}\right)\equiv 0\left(\mathrm{mod}q_{1}q_{2}...q_{n}\right)$$This means that there is another rendezvous point in addition to the i=j=0 point. It indicates
$$\sum_{\alpha =0}^{xN-1}h\left(s_{\alpha ,\beta^{*}}^{\lambda_{1}},s_{\alpha +v,<\beta^{*}+h{>}_{N}}^{\lambda_{2}}\right)\geq 2$$By Equations (2) and (3) for any $\beta^{*}\in \left[0,xN-1\right]$, we also have
(7)$$H_{min}\left(M\right)\geq 2$$It is shown that there are at least two rendezvous points in a column. □

**Example** **1.**
*x=2 and N=10.*

*Step 1: Choose γ=3 the largest multiplicative order ord(γ)=4 in $Z_{10}^{*}$.*

*Step 2: Let λ=0, we can generate an N×N matrix $S^{0}$ according to Equation (Equation 1).*

*Step 3: String the matrix $S^{0}$ 2 times to create the N×2N CHS $M^{0}$ as follows:*

$$\left[\begin{array}{cccccccccccccccccccc} 0 & 1 & 2 & 3 & 4 & 5 & 6 & 7 & 8 & 9 & 0 & 1 & 2 & 3 & 4 & 5 & 6 & 7 & 8 & 9\\ 1 & 2 & 3 & 4 & 5 & 6 & 7 & 8 & 9 & 0 & 1 & 2 & 3 & 4 & 5 & 6 & 7 & 8 & 9 & 0\\ 2 & 3 & 4 & 5 & 5 & 7 & 8 & 9 & 0 & 1 & 2 & 3 & 4 & 5 & 6 & 7 & 8 & 9 & 8 & 1\\ 3 & 4 & 5 & 6 & 6 & 8 & 9 & 0 & 1 & 2 & 3 & 4 & 5 & 6 & 7 & 8 & 9 & 0 & 7 & 2\\ 4 & 5 & 6 & 7 & 7 & 9 & 0 & 1 & 2 & 3 & 4 & 5 & 6 & 7 & 8 & 9 & 0 & 1 & 6 & 3\\ 5 & 6 & 7 & 8 & 8 & 0 & 1 & 2 & 3 & 4 & 5 & 6 & 7 & 8 & 9 & 0 & 1 & 2 & 5 & 4\\ 6 & 7 & 8 & 9 & 9 & 1 & 2 & 3 & 4 & 5 & 6 & 7 & 8 & 9 & 0 & 1 & 2 & 3 & 4 & 5\\ 7 & 8 & 9 & 0 & 0 & 2 & 3 & 4 & 5 & 6 & 7 & 8 & 9 & 0 & 1 & 2 & 3 & 4 & 3 & 6\\ 8 & 9 & 0 & 1 & 1 & 3 & 4 & 5 & 6 & 7 & 8 & 9 & 0 & 1 & 2 & 3 & 4 & 5 & 2 & 7\\ 9 & 0 & 1 & 2 & 2 & 4 & 5 & 6 & 7 & 8 & 9 & 0 & 1 & 2 & 3 & 4 & 5 & 6 & 7 & 8 \end{array}\right]$$


*Then we constructed a CHS $M^{1}$ with λ=1:*

$$\left[\begin{array}{cccccccccccccccccccc} 3 & 6 & 9 & 2 & 5 & 8 & 1 & 4 & 7 & 0 & 3 & 6 & 9 & 2 & 5 & 8 & 1 & 4 & 7 & 0\\ 6 & 9 & 2 & 5 & 8 & 1 & 4 & 7 & 0 & 3 & 6 & 9 & 2 & 5 & 8 & 1 & 4 & 7 & 0 & 3\\ 9 & 2 & 5 & 8 & 1 & 4 & 7 & 0 & 3 & 6 & 9 & 2 & 5 & 8 & 1 & 4 & 7 & 0 & 3 & 6\\ 2 & 5 & 8 & 1 & 4 & 7 & 0 & 3 & 6 & 9 & 2 & 5 & 8 & 1 & 4 & 7 & 0 & 3 & 6 & 9\\ 5 & 8 & 1 & 4 & 7 & 0 & 3 & 6 & 9 & 2 & 5 & 8 & 1 & 4 & 7 & 0 & 3 & 6 & 9 & 2\\ 8 & 1 & 4 & 7 & 0 & 3 & 6 & 9 & 2 & 5 & 8 & 1 & 4 & 7 & 0 & 3 & 6 & 9 & 2 & 5\\ 1 & 4 & 7 & 0 & 3 & 6 & 9 & 2 & 5 & 8 & 1 & 4 & 7 & 0 & 3 & 6 & 9 & 2 & 5 & 8\\ 4 & 7 & 0 & 3 & 6 & 9 & 2 & 5 & 8 & 1 & 4 & 7 & 0 & 3 & 6 & 9 & 2 & 5 & 8 & 1\\ 7 & 0 & 3 & 6 & 9 & 2 & 5 & 8 & 1 & 4 & 7 & 0 & 3 & 6 & 9 & 2 & 5 & 8 & 1 & 4\\ 0 & 3 & 6 & 9 & 2 & 5 & 8 & 1 & 4 & 7 & 0 & 3 & 6 & 9 & 2 & 5 & 8 & 1 & 4 & 7 \end{array}\right]$$


*and the CHS $M^{1}$ with cyclic shift τ=1(h=1,v=0):*

$$\left[\begin{array}{cccccccccccccccccccc} 7 & 3 & 6 & 9 & 2 & 5 & 8 & 1 & 4 & 7 & 0 & 3 & 6 & 9 & 2 & 5 & 8 & 1 & 4 & 7\\ 0 & 6 & 9 & 2 & 5 & 8 & 1 & 4 & 7 & 0 & 3 & 6 & 9 & 2 & 5 & 8 & 1 & 4 & 7 & 0\\ 3 & 9 & 2 & 5 & 8 & 1 & 4 & 7 & 0 & 3 & 6 & 9 & 2 & 5 & 8 & 1 & 4 & 7 & 0 & 3\\ 6 & 2 & 5 & 8 & 1 & 4 & 7 & 0 & 3 & 6 & 9 & 2 & 5 & 8 & 1 & 4 & 7 & 0 & 3 & 6\\ 9 & 5 & 8 & 1 & 4 & 7 & 0 & 3 & 6 & 9 & 2 & 5 & 8 & 1 & 4 & 7 & 0 & 3 & 6 & 9\\ 2 & 8 & 1 & 4 & 7 & 0 & 3 & 6 & 9 & 2 & 5 & 8 & 1 & 4 & 7 & 0 & 3 & 6 & 9 & 2\\ 5 & 1 & 4 & 7 & 0 & 3 & 6 & 9 & 2 & 5 & 8 & 1 & 4 & 7 & 0 & 3 & 6 & 9 & 2 & 5\\ 8 & 4 & 7 & 0 & 3 & 6 & 9 & 2 & 5 & 8 & 1 & 4 & 7 & 0 & 3 & 6 & 9 & 2 & 5 & 8\\ 1 & 7 & 0 & 3 & 6 & 9 & 2 & 5 & 8 & 1 & 4 & 7 & 0 & 3 & 6 & 9 & 2 & 5 & 8 & 1\\ 4 & 0 & 3 & 6 & 9 & 2 & 5 & 8 & 1 & 4 & 7 & 0 & 3 & 6 & 9 & 2 & 5 & 8 & 1 & 4 \end{array}\right]$$


*the CHS $M^{1}$ with cyclic shift τ=2(h=2,v=0):*

$$\left[\begin{array}{cccccccccccccccccccc} 4 & 7 & 3 & 6 & 9 & 2 & 5 & 8 & 1 & 4 & 7 & 0 & 3 & 6 & 9 & 2 & 5 & 8 & 1 & 4\\ 7 & 0 & 6 & 9 & 2 & 5 & 8 & 1 & 4 & 7 & 0 & 3 & 6 & 9 & 2 & 5 & 8 & 1 & 4 & 7\\ 0 & 3 & 9 & 2 & 5 & 8 & 1 & 4 & 7 & 0 & 3 & 6 & 9 & 2 & 5 & 8 & 1 & 4 & 7 & 0\\ 3 & 6 & 2 & 5 & 8 & 1 & 4 & 7 & 0 & 3 & 6 & 9 & 2 & 5 & 8 & 1 & 4 & 7 & 0 & 3\\ 6 & 9 & 5 & 8 & 1 & 4 & 7 & 0 & 3 & 6 & 9 & 2 & 5 & 8 & 1 & 4 & 7 & 0 & 3 & 6\\ 9 & 2 & 8 & 1 & 4 & 7 & 0 & 3 & 6 & 9 & 2 & 5 & 8 & 1 & 4 & 7 & 0 & 3 & 6 & 9\\ 2 & 5 & 1 & 4 & 7 & 0 & 3 & 6 & 9 & 2 & 5 & 8 & 1 & 4 & 7 & 0 & 3 & 6 & 9 & 2\\ 5 & 8 & 4 & 7 & 0 & 3 & 6 & 9 & 2 & 5 & 8 & 1 & 4 & 7 & 0 & 3 & 6 & 9 & 2 & 5\\ 8 & 1 & 7 & 0 & 3 & 6 & 9 & 2 & 5 & 8 & 1 & 4 & 7 & 0 & 3 & 6 & 9 & 2 & 5 & 8\\ 1 & 4 & 0 & 3 & 6 & 9 & 2 & 5 & 8 & 1 & 4 & 7 & 0 & 3 & 6 & 9 & 2 & 5 & 8 & 1 \end{array}\right]$$


*the CHS $M^{1}$ with cyclic shift τ=8(h=8,v=0):*

$$\left[\begin{array}{cccccccccccccccccccc} 6 & 9 & 2 & 5 & 8 & 1 & 4 & 7 & 3 & 6 & 9 & 2 & 5 & 8 & 1 & 4 & 7 & 0 & 3 & 6\\ 9 & 2 & 5 & 8 & 1 & 4 & 7 & 0 & 6 & 9 & 2 & 5 & 8 & 1 & 4 & 7 & 0 & 3 & 6 & 9\\ 2 & 5 & 8 & 1 & 4 & 7 & 0 & 3 & 9 & 2 & 5 & 8 & 1 & 4 & 7 & 0 & 3 & 6 & 9 & 2\\ 5 & 8 & 1 & 4 & 7 & 0 & 3 & 6 & 2 & 5 & 8 & 1 & 4 & 7 & 0 & 3 & 6 & 9 & 2 & 5\\ 8 & 1 & 4 & 7 & 0 & 3 & 6 & 9 & 5 & 8 & 1 & 4 & 7 & 0 & 3 & 6 & 9 & 2 & 5 & 8\\ 1 & 4 & 7 & 0 & 3 & 6 & 9 & 2 & 8 & 1 & 4 & 7 & 0 & 3 & 6 & 9 & 2 & 5 & 8 & 1\\ 4 & 7 & 0 & 3 & 6 & 9 & 2 & 5 & 1 & 4 & 7 & 0 & 3 & 6 & 9 & 2 & 5 & 8 & 1 & 4\\ 7 & 0 & 3 & 6 & 9 & 2 & 5 & 8 & 4 & 7 & 0 & 3 & 6 & 9 & 2 & 5 & 8 & 1 & 4 & 7\\ 0 & 3 & 6 & 9 & 2 & 5 & 8 & 1 & 7 & 0 & 3 & 6 & 9 & 2 & 5 & 8 & 1 & 4 & 7 & 0\\ 3 & 6 & 9 & 2 & 5 & 8 & 1 & 4 & 0 & 3 & 6 & 9 & 2 & 5 & 8 & 1 & 4 & 7 & 0 & 3 \end{array}\right]$$


*The elements in red represent the rendezvous points in the CHS $M^{0}$ and $M^{1}$.*


**Theorem** **2.**
*For any integer N≥3, the MTTR and MFTTR of the CHS set generated by Construction 1 are $\frac{xN^{2}}{2}$ and $\frac{xN^{2}}{2}$, respectively.*


**Proof.** According to Theorem 1, there are at least two rendezvous in a column if $\gamma^{\lambda_{1}}-\gamma^{\lambda_{2}}=\frac{N}{q_{u}}$. Thus, we can have the following discussion:Let *k* be an integer; *k* denotes the total number of columns with rendezvous points. *h* is denoted as the horizontal shift, then we have
$$k=\left\{\begin{array}{c} xN-h,\mathrm{if}\;h\;\mathrm{is}\;\mathrm{odd}\\ h,\mathrm{if}\;h\;\mathrm{is}\;\mathrm{even} \end{array}\right.$$Let r(α,β) represent the coordinate of the rendezvous point, where α corresponds to the row index, and β corresponds to the column index. If $r_{1}\left(\alpha_{1},\beta_{1}\right)$ and $r_{2}\left(\alpha_{2},\beta_{2}\right)$ are two rendezvous points then
$$MTTR=\left|\alpha_{2}-\left(\alpha_{1}-1\right)\right|\times xN-\beta_{1}-\left(xN-\beta_{2}\right)$$
$$MFTTR=\alpha_{1}\times \beta_{1}$$
(1)If $0<k\leq \frac{N}{2}$:If k=1, at this time the rendezvous point appears only in the last column (h=xN−1). Consider the worst situation where $r_{1}\left(\alpha_{1}=\frac{N}{2},\beta_{1}=xN\right)$, $r_{2}\left(\alpha_{2}=N,\beta_{2}=xN\right)$; therefore, MTTR = $\left|\alpha_{2}-\left(\alpha_{1}-1\right)\right|\times xN-\beta_{1}-\left(xN-\beta_{2}\right)$ = $\left(N-\frac{N}{2}+1\right)\times xN-xN-\left(xN-xN\right)$ = $\frac{xN^{2}}{2}$, and MFTTR = $\alpha_{1}\times \beta_{1}$=$xN\times \frac{N}{2}$ = $\frac{xN^{2}}{2}$.(2)If $\frac{N}{2}<k\leq xN-1$:Since each row satisfies at least one rendezvous, the maximum MTTR is generated when only one rendezvous happens between both rows, i.e., $r_{3}\left(\alpha_{3},\beta_{3}\right)$, $r_{4}\left(\alpha_{3}+1,\beta_{3}-1\right)$. MTTR = $\left|\alpha_{3}+1-\left(\alpha_{3}-1\right)\right|\times xN-\beta_{3}-\left(xN-\beta_{3}+1\right)$ = xN−1 is generated at this time. However, for r(α=1,β=xN) (the last column of the first row), it appears that MFTTR = α×β = 1×xN = xN.
According to the above analysis, we have:
$$MTTR=\left\{\begin{array}{c} \frac{xN^{2}}{2},0<k\leq \frac{N}{2}\\ xN-1,\frac{N}{2}<k\leq xN-1 \end{array}\right.$$
$$MFTTR=\left\{\begin{array}{c} \frac{xN^{2}}{2},0<k\leq \frac{N}{2}\\ xN,\frac{N}{2}<k\leq xN-1 \end{array}\right.$$□

**Example** **2.**
*When N=10, $L=2N^{2}=200$, the MTTR and MFTTR performance of the new construction is shown below:*

$$MTTR=\left\{\begin{array}{c} 100,0<k\leq \;5\\ 19,5<k\leq \;19 \end{array}\right.$$


$$MFTTR=\left\{\begin{array}{c} 100,0<k\leq \;5\\ 20,5<k\leq \;19 \end{array}\right.$$


*Therefore, for N=10, MTTR=100 and MFTTR=100.*


**Corollary** **1.**
*For the new construction when N=p, and p is a prime, MTTR and MFTTR take xp−1 and xp, respectively, for any integer p≥3.*


**Proof.** In any combination of vertical and horizontal circular shifts, the new construction always has at least one rendezvous in each column between any pair of p×xp CH matrices.When N=p and *p* is a prime, MTTR = xp−1 because each row has xp elements, and each column consists of different factors. The worst case of the first rendezvous occurs in the last column of the CH matrix so that MFTTR = xp when all licensed channels are available. □

## 4. Performance Comparison

In this section, we compare the MTTR, the TTR mean and the variance of the new construction with S-ACH [13] and E-AHW [14] under the same channel *N*. Since the new construction is a flexible parameter, in order to accurately compare the three constructions, Figure 3, Figure 4 and Figure 5 illustrate the constant new construction of length x=2. The performance of full DoR is not considered in S-ACH [13] and E-AHW [14], so we focus on their MTTR performances here. Thus, the new construction has MTTR = $N^{2}$, where *N* is non-prime, while S-ACH [13] and E-AHW [14] have MTTR of $6lN^{2}-N+1$ and 3(l+1)Np−N+1, respectively. Here, *l* stands for the ID sequence length and *p* is the smallest prime number greater than *N*.

Figure 3 compares the new construction with the S-ACH [13] and E-AHW [14] construction for MTTRs with non-prime *N*. Observe that the MTTR deteriorates with increasing *N*. This is due to the rise in the licensed channel capacity, which leads to an increase in the distance between channel convergence communications. In the figure, we observe a concentration of convertible channels when the channel capacity increases. This is because, under asynchronous conditions, the ID expansion sequence affects both S-ACH [13] and E-AHW [14] construction, making the distribution between the converging channels uneven. However, the ID sequences do not affect the new construction, resulting in a better MTTR = $N^{2}\left(x=2\right)$ performance.

Figure 4 and Figure 5 compare the TTR mean and the variance between the construction with the same parameters as in Figure 3. For each construction, the TTR mean is calculated by averaging all TTRs in all simulation iterations, and the TTR variance is calculated by averaging over all TTRs. It can be seen in Figure 4 and Figure 5 that the mean and variance of the TTR for S-ACH [13] and E-AHW [14] are unstable, which is due to uneven distributions caused by convergent channel aggregation. Additionally, the new construction’s TTR mean value is not the smallest among them. However, a tiny TTR variance demonstrates that the new construction of channel rendezvous is evenly distributed, and the channel rendezvous is not concentrated in a specific time slot, which makes the communication quality more reliable.

## 5. Conclusions

In this paper, a set of asynchronous channel hopping sequences with new parameters is proposed based on arbitrary licensed channels. It is used to implement a cognitive radio network for asynchronous communication. The novelty of our approach lies in the fact that the new construction uses interleaving techniques to construct a family of channel-hopping sequences with new parameters, allowing it to be applied in different scenarios. Rendezvous is guaranteed to occur between any two sequences, and the rendezvous time is unpredictable and inhomogeneous. The simulated results show that the new construction is more balanced between the MTTR and small TTR mean and variance. In wireless communication, adopting this technology will be helpful in numerous communication scenarios. It helps reduce the channel congestion and interference that may jeopardize the dependability and security of communication. In future work, we will investigate new CHSs to achieve better performances of DoR, MTTR and MFTTR.

## Figures and Tables

**Figure 1 entropy-25-01473-f001:**
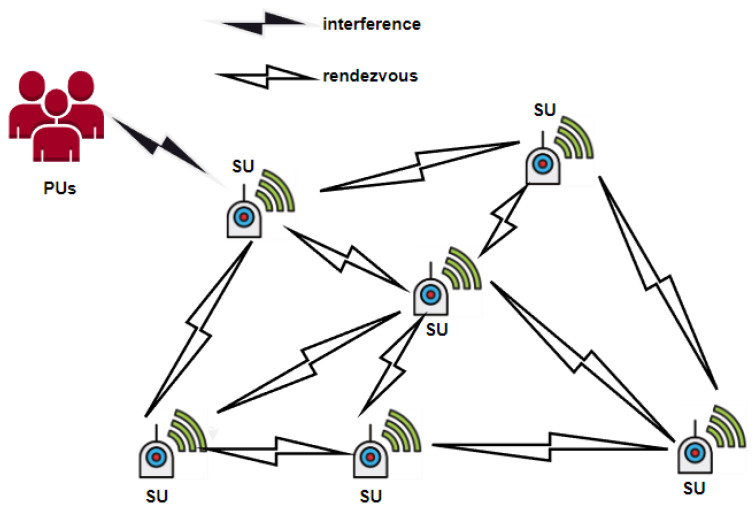
An illustration of a rendezvous between SUs based on distribution.

**Figure 2 entropy-25-01473-f002:**
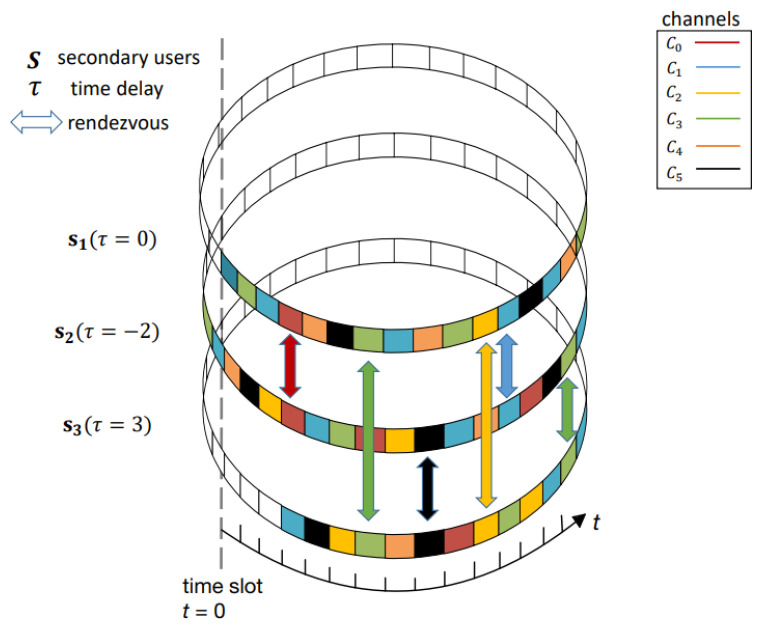
Example of SUs asynchronous rendezvous model.

**Figure 3 entropy-25-01473-f003:**
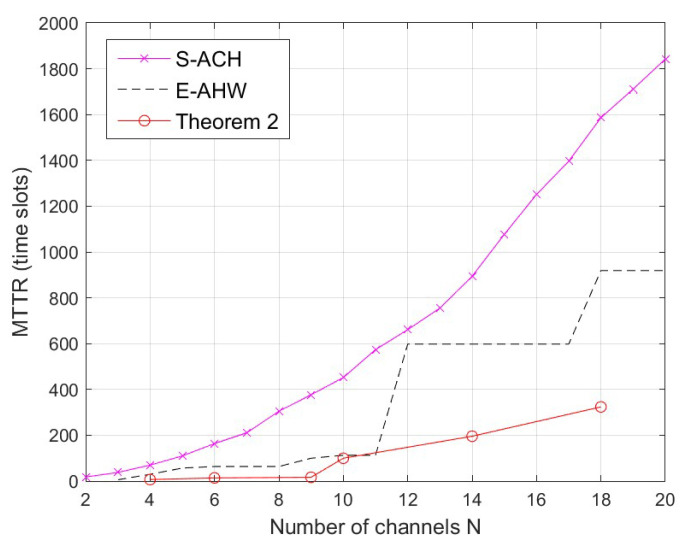
Comparison of the MTTR of related constructions under the same channel *N*.

**Figure 4 entropy-25-01473-f004:**
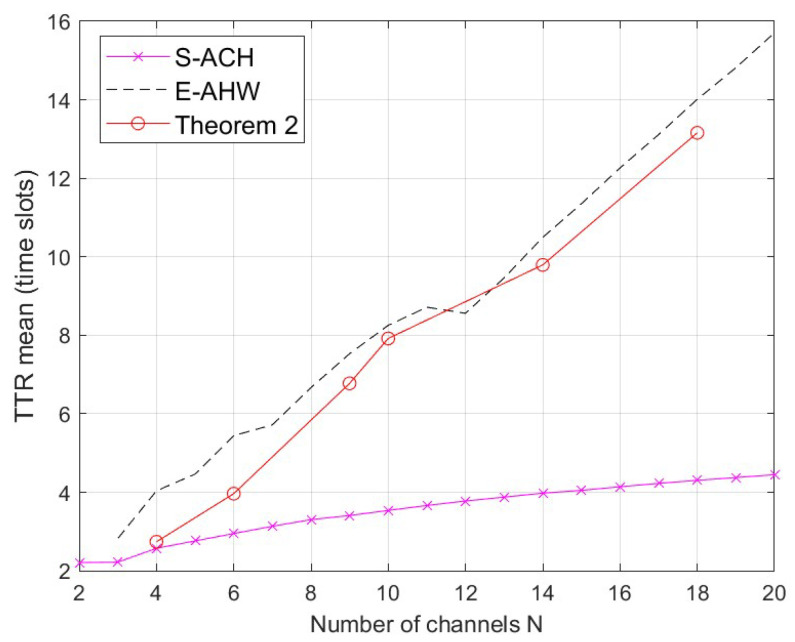
Comparison of TTR mean of the correlation construction under the same channel *N*.

**Figure 5 entropy-25-01473-f005:**
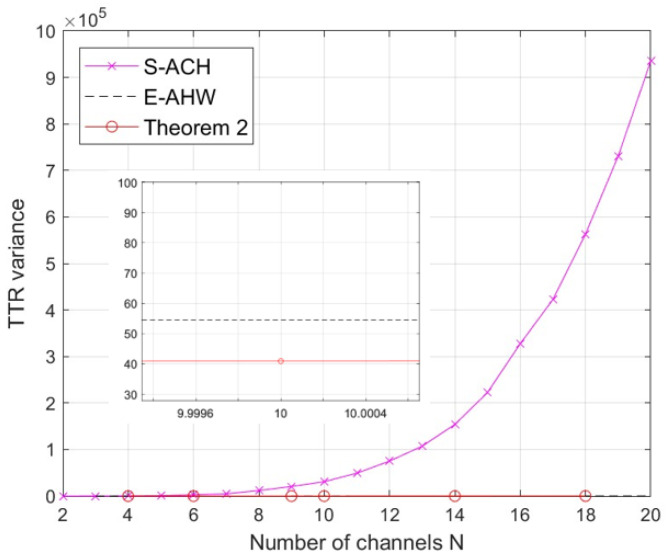
Comparison of TTR variance of the correlation construction under the same channel *N*.

**Table 1 entropy-25-01473-t001:** Comparison of Related CHSs.

	Period *L*	MTTR	MFTTR
JS [12]	$3Np^{2}$	4p	3p
S-ACH [13]	$6lN^{2}$	$6lN^{2}-N+1$	$6lN^{2}$
SQCH [9]	$2N^{3}+N^{2}$	$2N^{2}+N$	$2N^{3}+N^{2}$
E-AHW [14]	3(l+1)Np	3(l+1)Np−N+1	3(l+1)Np
DSCR [10]	$2p^{2}+p\left\lfloor p/2\right\rfloor$	$2p^{2}+p\left\lfloor p/2\right\rfloor -p+1$	$2p^{2}+p\left\lfloor p/2\right\rfloor$
ACHPS [15]	$p^{2}$	$p^{2}-p$	$p^{2}$
IDEAL-CH [16]	$2p^{2}$	$2p^{2}-p+1$	$2p^{2}$
CM2P-CH [11]	$2p^{2}$	2p	2p
Theorem2	$xN^{2}$(N≠p)	$\frac{xN^{2}}{2}$	$\frac{xN^{2}}{2}$
Corollary1	$xp^{2}$(N=p)	xp−1	xp

*N* and *x* denote positive integers, *p* is a prime, and *l* represents the length of the ID sequence.

## Data Availability

Not applicable.

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
