# Peer review of "New Construction of Asynchronous Channel Hopping Sequences in Cognitive Radio Networks"

_entropy, 2023, doi:10.3390/e25101473_

Round 1

Reviewer 1 Report

Moderate editing of English language required

Author Response

See the pdf

Reviewer 2 Report

In this paper, the authors claim that they construct a set of variable-length asynchronous hopping sequences with any licensed channel. There are several concerns shown as follows:

1. What do the authors mean on the terminology "variable-length"? Does it just mean that the proposed algorithm can be constructed by xN^2 with arbitrary integer x? For the variable-length asynchronous hopping sequences, it should be referred to the users can use different length of hopping sequences.

2. What is advantage to use the larger x? In the authors' examples, can x=1? 

3. In the construction algorithm shown by the authors, each matrix should be the same. However, the example in page 5, the left matrix is different  from the right matrix.

4. All the proofs are too rough, especially the proof in Theorem 2. It looks like a statement instead of a proof. 

5. The paragraphs in "Section 2 Related work" should be revised to clearly describe the contributions of the papers.

The paragraphs in "Section 2 Related work" should be revised to clearly describe the contributions of the papers.

Author Response

See the pdf

Round 2

Reviewer 2 Report

I think the paper can be accepted by mentioning the following things in the abstract and contributions:

1. The MTTR improvement by 1 is sacrificed by reducing the full DoR to  DoR to DoR less than N, where N is the number of channels. To compare MTTR without mentioning DoR is not a fair comparison.

2. Please mention how many of CH sequences can be constructed for given N.

3. In general, for given N, the goal for CH sequence design is to find the smallest length to support to desirable features. The authors just concatenate the CH matrices to adjust the length of CH sequences which is trivial in general.

I think the paper can be accepted by mentioning the following things in the abstract and contributions:

1. The MTTR improvement by 1 is sacrificed by reducing the full DoR to  DoR to DoR less than N, where N is the number of channels. To compare MTTR without mentioning DoR is not a fair comparison.

2. Please mention how many of CH sequences can be constructed for given N.

3. In general, for given N, the goal for CH sequence design is to find the smallest length to support to desirable features. The authors just concatenate the CH matrices to adjust the length of CH sequences which is trivial in general.

Author Response

Please see the PDF for a review comment responses.

Round 3

Reviewer 2 Report

I agree on the changes the authors made.

I agree on the changes the authors made.